# How Social is It? A Benchmark for LLMs' Capabilities in Multi-user Multi-turn Social Agent Tasks

## Abstract

Expanding the application of large language models (LLMs) to societal life, instead of primary function only as auxiliary assistants to communicate with only one person at a time, necessitates LLMs' capabilities to independently play roles in multi-user, multi-turn social agent tasks within complex social settings. However, currently the capability has not been systematically measured with available benchmarks. To address this gap, we first introduce an agent task leveling framework grounded in sociological principles. Concurrently, we propose a novel benchmark, How Social Is It (we call it HSII below), designed to assess LLM's social capabilities in comprehensive social agents tasks and benchmark representative models. HSII comprises four stages: format parsing, target selection, target switching conversation, and stable conversation, which collectively evaluate the communication and task completion capabilities of LLMs within realistic social interaction scenarios dataset, HSII-Dataset. The dataset is derived step by step from news dataset. We perform an ablation study by doing clustering to the dataset. Additionally, we investigate the impact of chain of thought (COT) method on enhancing LLMs' social performance. Since COT cost more computation, we further introduce a new statistical metric, COT-complexity, to quantify the efficiency of certain LLMs with COTs for specific social tasks and strike a better trade-off between measurement of correctness and efficiency. Various results of our experiments demonstrate that our benchmark is well-suited for evaluating social skills in LLMs.

## 1 Introduction

Large language models (LLMs) enhance their expressive and reasoning capabilities through an increase in model parameters, depth, and breadth. They exhibit robust knowledge retention and reasoning abilities, and are continuously evolving. Recent surveys (Zhao et al., 2024), (Minaee et al., 2024), and (Gao et al., 2023b) provide comprehensive and detailed insights into this evolution. In practical applications, LLMs have shown significant potential across various domains, contributing notably to multi-agent systems (Han et al., 2024), (Guo et al., 2024a), (He et al., 2024), digital humans (Yang et al., 2024), (Zhang et al., 2023), embodied intelligence (Li et al., 2024b), (Song et al., 2023), education, intelligent customer service (Xu et al., 2024), (Shi et al., 2024), and code generation (Jiang et al., 2024), (Hassid et al., 2024), bringing artificial intelligence closer to everyday life. However, when compared to internet technology, which has become ubiquitous in social interactions through iterative development, there remains a more discernible gap between LLMs developer and non-developer accessibility. For instance, LLMs usually struggle to communicate independently with customers without supervision and do not excel in roles such as daily butler services or managing comprehensive company operations beyond simple tasks. Beyond the underutilization of current computational power, a significant reason may be the lack of capabilities in LLMs to independently and skillfully interact in complex social scenarios. To examine those possibilities we need to do precise evaluation to social capabilities.

To study LLMs' capabilities in complex social tasks is also essential for enhancing LLM sociology analysis. Recent exploration of the rationality and biological traits of LLMs, as discussed in Chen et al. (2023c) and Lyu et al. (2024), has been complemented by research simulating virtual societies

and systems through dialogues among multiple LLMs. This research aims to analyze their social attributes and perform a social division of labor, as exemplified in works such as Gurcan (2024), Dai et al. (2024), and Gao et al. (2023a). These endeavors, however, may be constrained by their idealized scenario settings, which limit the degree of realism (Zhou et al., 2024a). By building more complex and closer-to-reality agent tasks in social scenes based on sociology theory, and then benchmarking above them we may get sociology about LLM more solid.

Up to date the significance of LLMs' interpersonal communication skills has become gradually recognized, but current benchmarks have not fully covered this. To evaluate these skills, some works such as SOTOPIA-EVAL (Zhou et al., 2024b) and MUCA (Mao et al., 2024) have been made. SOTOPIA-EVAL focuses on designing scenarios to assess social intelligence through role-playing, comparing models against human performance. MUCA, on the other hand, simulates group interactions to establish a framework for determining chat targets' interactions with specified objects. Both works highlight the importance of multi-user dialogue in social relationships and the need for evaluation. Yet, currently there has been no work to bridge social dialogue scenarios with traditional dyadic dialogue assessments, explore their interrelation from a sociological perspective, and build a systematical benchmark for overall evaluation in all social capability dimensions.Moreover, mainstream evaluation frameworks, including thep (Chen et al., 2024) arena (Chiang et al., 2024), GPQA (Rein et al., 2023), and security assessment frameworks (Zhang et al., 2024c) (Li et al., 2024a), also do not explicitly assess social communication capabilities as a distinct dimension alongside mathematical, coding, and other critical thinking skills.

As our first approach to assess social competencies effectively, it is imperative to dissect the foundational elements of social interaction through a sociological lens and reinterpret them within the framework of LLMs. Our approach is bolstered by seminal sociological works that analyze social dynamics (Goffman, 1959; Duncan, 1972; Clark & Brennan, 1991), alongside contemporary investigations into LLMs from a sociological standpoint (Dai et al., 2024; Lan et al., 2024). Furthermore, the field of artificial intelligence, especially multi-agent systems, has provided valuable insights into the analysis and reconstruction of hierarchical systems (Li et al., 2019). By integrating these perspectives, we introduce a tiered division of tasks for social agents, categorizing them into four distinct levels: the first being fundamental and well-explored, the subsequent two being relatively autonomous, and the final level representing an integration of the former two.

With novel LLM sociology framework We endeavor to develop high-caliber evaluation datasets akin to established benchmarks. Deviating from the common practice of repurposing existing LLM datasets, we opt to initiate our dataset construction with manually and fairly reviewed-filtered real news data. The news data is algorithmically clustered and detoxified to capture a more authentic and representative cross-section of real-world social scenarios. Leveraging the comprehension and summarization prowess of the GPT4 model (OpenAI et al., 2024), we refine this data further. Subsequently, human evaluators curate and amend the data based on predefined criteria, yielding a collection of social scenarios featuring multiple participants and a spectrum of conflicts. The presence of heightened conflict is instrumental in rigorously testing the models' capabilities within intricate social contexts. This process culminates in the creation of a multi-user multi-turn dialogue dataset that is intrinsically linked to these scenarios.

In this study, we delve into the nature of evaluation methodologies for models in multi-user multi-turn social tasks and propose a novel metric HSII score. A recent study (Ren et al., 2024) introduces a framework aimed at enhancing social norms. Furthermore, Mao et al. (Mao et al., 2024) emphasize the importance of selecting an interlocutor and crafting dialogue content in multi-turn conversations. This involves determining "who to converse with and what to convey," while also establishing an interaction pattern across multiple users through extended dialogues. Building on these insights, we have designed a four-tiered evaluation protocol: prompting the model to respond with certain format, enabling the model to select from a broader array of potential interlocutors within our curated multi-turn dialogue dataset, articulating transitions between turns, and ensuring the continuity and stability of the dialogue post-switch, all of whose results was compiled up to compute the final HSII score. In this track we carry out our experiments on LLMs and propose our discoveries.

Additionally, when talking about capabilities in certain social scenes, one may question how Chain of Thought (COT) (Wei et al., 2022) affect LLMs' performance, given the potential of COT to augment social interaction skills within social contexts, iterative reasoning loops Qin & Cong (2023), and scholarly work suggesting that a well-crafted COT can address high-level challenges in a math-

emetical framework Zhang et al. (2024b). Hence in our benchmark we introduce another novel metric, COT complexity of LLMs under specific COT configurations. By assessing the minimum number of reasoning cycles a model must undergo under a thoughtfully crafted set of COT within the model's self-reflection to strike given accuracy threshold, we effectively benchmark the cognitive efficiency of various models.

We summarize our contributions in two main folds:

1. We make investigation about more complex tasks in social life and propose a systematical formulation of multi-user multi-turn social tasks structure.

2. We introduce the How Social Is It (HSII), a statistical metric for quantifying social capability in multi-user multi-turn complex task scenes based on theoretical derivation and sociological conclusions. We then present how the dataset construction and evaluation pipeline is extracted in detail.

3. We introduce the COT complexity metric to measure how efficient LLMs are to do reasoning and reflection along given set of COTs to meet given standard. We develop a pratical pipeline for this evaluation.

## 2    RELATED WORK

**Social Relationship and Social Scene.** Crafting social scenarios and interactions involves several critical elements and stages. Social scenarios are intricately woven from components such as setting, participants, and behavioral norms(Goffman, 1959). The process of social interaction generally unfolds through stages including initiation, development, and termination(Duncan, 1972). Dialogue acts involve the selection of conversational targets and the management of transitions between different speakers(Clark & Brennan, 1991). Navigating multi-turn dialogues with a single conversational target necessitates the application of adjacency pairs and the preservation of topical coherence(Clark & Schaefer, 1987).

**Agent Task Stratification.** Within the academic community, significant progress has been made in the field of agent task stratification. This process involves the systematic breakdown of complex missions into smaller, more tractable subtasks, which is essential for the effective operation of Multi-Agent Systems (MAS). For instance, Hierarchical Reinforcement Learning (HRL)(Li et al., 2019) tackles the challenges posed by sparse rewards and complex environments by decomposing intricate tasks into simpler subtasks. Meta-Task Planning (MTP)(Zhang et al., 2024a), a strategy for simplifying complex task planning in collaborative, LLM-based MAS, furthers this approach by decomposing tasks into a sequence of subordinate tasks, or meta-tasks, which are then translated into actionable steps. AI Agents can be classified into a spectrum of levels, ranging from Level 0 (non-AI, basic tools) to Level 5 (highly advanced agents exhibiting personality and cooperative interaction)(Huang, 2024). Each ascending level integrates additional modules and functionalities, thereby augmenting the AI capabilities and utility of the agents. The SMART-LLM framework(Kannan et al., 2023) exemplifies this progression, applying LLMs to multi-robot task planning. It translates high-level directives into actionable multi-robot plans through a systematic process that includes task decomposition, coalition formation, and task allocation.

**LLM Intelligent Agent Application Evaluation and Benchmarking.** Evaluating large AI models necessitates a rigorous methodology to assess their performance, robustness, and reliability(NIST, 2022). This evaluation is essential for guaranteeing that models adhere to benchmarks of safety, efficacy, and ethical standards prior to their operational deployment(Committee, 2024). Prominent benchmarks for LLM assessment include ImageNet(Russakovsky et al., 2015), a seminal benchmark for computer vision models, GLUE(Wang et al., 2019a), which evaluates natural language understanding, and ArenaBench(Kastner et al., 2022), a comprehensive benchmark designed to assess AI systems across a spectrum of tasks and environments(MLSys, 2020). These benchmarks are instrumental in promoting transparency and propelling the evolution of AI technology.

## 3    PRELIMINARIES

The primary focus lies in evaluating model efficacy in social tasks characterized by multi-turn dialogues within intricate scenarios, involving numerous conversational targets. In this study, we rigor-

ously assess the performance of models on originally built multi-user multi-turn social task datasets HSII. These tasks necessitate a diverse array of capabilities and often entail intricate interactions among multiple participants, underscoring the importance of considering both factors.

**Social Task capability Objectives Division.** (Zhou et al., 2024b) designs a multi-dimensional framework with various objectives, including the following and so on:

- **Goal Completion (GOAL)** This is the extent to which the agent achieves their goals.

- **Believability (BEL)** This focuses on the extent to which the agent's behavior is perceived as natural, realistic, and aligned with the agent's character profile, thus simulating believable proxies of human behavior.

- **Knowledge (KNO)** This captures the agent's capability to actively acquire new information.

- **Secret (SEC) [-10-0]** This measures the need for agents (humans) to keep their secretive information or intentions private.

**Framework of Multi-User Chat (MUC)**The multi-user framework architecture and information flow consist of three major modules: Sub-topics Generator, which generates the initial sub-topics; Dialog Analyzer, which extracts short-term and long-term features from chat history; Utterance Strategies Arbitrator, which determines the dialog acts corresponding to our design dimensions.

Overall, the Sub-topics Generation is executed once, and the Dialog Analyzer and Utterance Strategies Arbitrator are executed sequentially for every next utterance, which ensures latency-efficiency in front of higher message traffic and complex interactions from multiple users. Among them, the Dialog Analyzer consists of sub-modules containing Sub-topic Status Update, Utterance Feature Extractor, Accumulative Summary Update, Participant Feature Extractor. The Utterance Strategies Arbitrator includes modules like Direct Chatting, Initiative Summarization, Participation Encouragement, Sub-topic Transition, Conflict Resolution.(Mao et al., 2024)

## 4 FRAMEWORK

Here, we delineate our social agent architecture, which is meticulously crafted.

Drawing inspiration from pivotal sociological theories that dissect social relationships (Abbott, 2020), (Bondarenko, 2020), and (Tromp & Vial, 2022), we dissect social interactions into three critical facets: **Object Transition** Identifying the subsequent conversational target during the object transition phase.**Transitional Utterance** Formulating and selecting dialogue content for the forthcoming interaction with the designated conversational target.**Post-Transition Multi-Turn and Multi-Level Dialogue** Engaging in multi-turn dialogues with the chosen conversational target post-transition, evaluating the aggregate effects and ultimate outcomes.

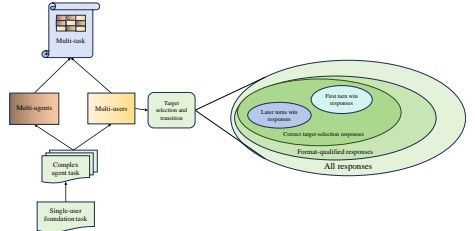

Figure 1: Main leveling of agent task and capability evaluation. On the left is different levels of social tasks including basic single-user tasks, multi-agents and multi-users tasks based on the first ones, and final multi-task ones on top. We mainly put sight on multi-user tasks. Then on the right is our four-step evaluation framework for multi-user tasks.

### 4.1 AGENT TASK LEVELING
FROM THE SOCIAL AGENT PERSPECTIVE

Here we give brief construction for two agent task groups. More precise inside structure for both types are presented in appendix part.

**Single-User Foundational Agent Tasks** Within a predefined protocol, one agent addresses inquiries from only one single user and execute API call commands with given tools. The goal is to attain

high precision in instruction following.The foundational approach entails a single-step agent call, while an alternative strategy is predicated on the Chain of Thought (CoT) (Wei et al., 2022).

**Complex Agent Tasks** Expanding upon the single-user foundational task, we define composite tasks as those encompass two categories of agent enhancement that may interrelate and nest: multi-agents and multi-users. Multi-agents tasks stand for that multiple LLM agents collaborate to jointly accomplish a single task (Han et al., 2024)(Guo et al., 2024b)(Wu et al., 2023).Multi-users tasks ensemble that single LLM agent serves multiple users, necessitating the determination of the current conversational target, the dialogue content to facilitate target transition, and the optimization of outcomes for all targets post-transition. The final comprehensive multi-tasks mean single LLM agent performs diverse tasks for multiple users, exemplifying generalizability Tan et al. (2023)(Chen et al., 2023b).

## 4.2 TALKING TARGET TRANSITION IN MULTI-USER MULTI-TURN DIALOGUE SYSTEMS

In practical applications, dialogue systems often encounter scenarios where a single agent must engage with multiple distinct users, each requiring tailored responses. This dynamic unfolds across multiple dialogue turns, with target transitions facilitating the switch between interlocutors. For instance, in a school setting like a parent-teacher conference, various targets like parents, students, teachers, principals are involved. Certain information, such as a student's report card, is restricted to specific parties. A straightforward solution is to have a single intelligent assistant handle these diverse needs simultaneously, representing the temporal relationships through a unified target. However, challenges arise in complex social scenarios where the needs of multiple parties are interdependent. For example, an intelligent assistant may need to interact with the parents of high-achieving students and teachers to glean effective learning strategies first, which can then inform advice for parents of struggling students. In such cases, the assistant must assess the priority of responding to different targets, drawing on the social dynamics of real-world conversations (Choi et al., 2020)(Peralta et al., 2022)(Adams et al., 2022), and generate utterances based on the unique information pertinent to the selected target. In our approach we focus on the evaluation of LLMs' performance in those complex scenes, separately in single-turn and with COT form, under different requirements.

## 5 METHODS

In this section, we introduce the pipeline to build our HSII dataset and then we propose HSII evaluation framework for LLMs' capability in multi-user multi-turn social agent tasks based on HSII dataset.

### 5.1 CONSTRUCTION OF A MULTI-USER MULTI-TURN DIALOGUE DATASET FROM NEWS DATASETS

Our approach begins with the strategic selection of one to two keywords to seed news searches, programmatically retrieving relevant news articles and documents(Leeb & Schölkopf, 2024) (Gao et al., 2024). These documents are then employed to craft thematic descriptions. Utilizing the thematic descriptions, we proceed to simulate dialogue data by meticulously extracting and organizing the pertinent thematic elements into scenario components with GPT4(OpenAI et al., 2024). In the final stage, these components are meticulously assembled to form comprehensive multi-user, multi-turn dialogues in HSII, by GPT4 and manual refining. To optimize the resource-intensive search process, we harness pre-processed offline news datasets as seeds for scenario generation. In our pipeline to augment the dataset's authenticity, we curate news report excerpts from diverse sources that encapsulate real-world events and distill the key details and logical connections within the reports, transforming them into a structured background setup with multiple fields, including domain, brief Scene Description, main Scene Participants and Social Relationships, and Potential Conflicts Among Participants.The entire process is graphically represented 2a for clarity.

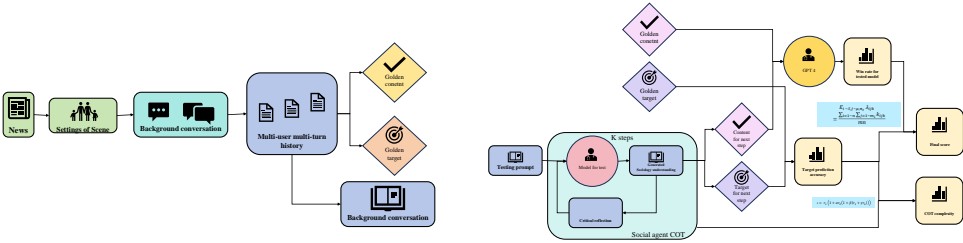

(a) Evaluation dataset construction design.  (b) HSII evaluation framework pipeline

Figure 2: Evaluation dataset construction design and HSII evaluation framework pipeline.

## 5.2 EVALUTION FRAMEWORK

### 5.2.1 THEORETICAL SETTINGS

**HSII overall score.** We introduce a novel metric, overall HSII score, to measure how social one tested LLM performs in four evaluation stages: verifying whether parsed response can suit requirements; selecting the next talking target in the social task scene; generating the first statement after switching; and engaging in sustained dialogue after switching, just as follows:

**Definition 1.** *For each test case $s_i$ in the test dataset $S$ with size $n$, we input it to the model $\pi_t$ and get response $\mu_i$. Then we parse $\mu_i$ to required pattern. Then we count the rate of successful parsing $r_1 = n_1/n$, then from parsed output dict we match target selection in this step $t_i$ with golden target $\boldsymbol{t}_i$ and count $n_2 = \sum_{i=1}^{n} 1(t_i = \boldsymbol{t}_i)$ to get successful target selection rate $r_2 = n_2/n_1$. We input first utterance content $\omega_i$ and golden one $\Omega_i$ to GPT4 for judgment which wins, loses or equals, on reverse sides to avoid positional bias, getting win rate of the test model $r_3 = n_3/n_2$. Finally we prompt testing model to chat for several turns. Similarly get long-run win rate $r_4 = n_4/n_2$. The final overall HSII score, noted as $\iota$, as*

$$\iota = r_1(1 + \alpha r_2(1 + \beta(r_3 + \gamma r_4))) \tag{1}$$

.

$\alpha, \beta, \gamma$ in the equation should be experimental hyper-parameters for overall evaluation. Here we take weight $\alpha = 1.0$, $\beta = 1.0$ and $\gamma = 1.0$ as equal for each stage for fairness. Apart from sociological background discussed earlier, approach of this metric ensures sufficient discrimination between similar test models. An overall analysis is provided in the appendix section.

**COT complexity.** Previous work proposes COT boost LLMs' performance(Wei et al., 2022). However it's simple that the COT methodology demands more computational resources than single-turn problem-solving. Also notably, longer COTs are computationally more intensive than their shorter counterparts. To quantitatively assess the efficiency of AI models, we introduce a nature metric: the social task complexity, as following. This metric evaluates a model's performance under specific COT designs when tackling certain questions.

**Definition 2.** *Given a test dataset $S$ comprising $n$ test cases $s_i$, we construct a standardized COT set $\boldsymbol{\mu} = \{\mu_{i1}, \mu_{i2}, \ldots, \mu_{im_i}\}$ for each test case $s_i$, with a set size of $m_i$. This COT set serves as a guide for various models $\boldsymbol{\pi} = \{\pi_1, \pi_2, \ldots, \pi_K\}$ to deliberate and respond to queries. For a particular model $\pi_t$, when it produces an answer aligning with the golden standard for $s_i$ under a specific COT $\mu_{ij}$ after $k_{ij}$ iterations of reflection and guidance, the COT complexity $\lambda_{ijt}$ for this social task $s_i$ under $\mu_{ij}$ for $\pi_t$ is recorded as $k_{ijt}$. In scenarios where the problem's complexity surpasses the capabilities of the current COT-framework-model pair, the COT complexity $\lambda_{ijt}$ is regraded as infinite.The COT complexity for model $\pi_t$ across the dataset $S$ is then defined as the average COT complexity under all test queries and corresponding COTs, mathematically expressed as:*

$$E_{i \sim S, j \sim \boldsymbol{\mu}, \pi_t} \lambda_{ijt} = \frac{\sum_{i=1}^{n} \sum_{j=1}^{m_i} k_{ijt}}{mn} \tag{2}$$

.

### 5.2.2 EVALUATION PIPELINE

In our proposed evaluation framework HSII, the multi-user dialogue capabilities of a LLM agent are rigorously assessed through both objective and subjective measures. Main evaluation pipeline is displayed in 2b.

The objective evaluation focuses on accuracy of target selection, which is quantified by calculating the proportion of correct next-target selections made by test LLM across all test cases. The subjective evaluation assesses the quality of the first-utterance and long-run statements generated by the model. Here we adopt the win rate metric introduced in ToolBench (Qin et al., 2023) to gauge the overall performance. There is one difference that we adopt both GPT4 and human-eval to get final win rate. Incorrect selections result in no score, as they lead to an invalid dialogue sequence by the LLM agent. But if in later phase the response is unfavorable, some score may still be awarded for correct selection. In appendix section we provide a rough theoretical foundation and sociology meaning for this approach.

## 6 EXPERIMENTS

### 6.1 EVALUATION DATASET BUILD

Utilizing the methodology outlined we construct HSII dataset with two steps. We begin with generating scenarios that encompass target transitions. By employing top-k scenario sampling and ascertaining which scenarios accurately meet predefined criteria and mirror real-world complexities involving intricate social dynamics and conflicts, we refine the scenarios and craft representative multi-user multi-turn dialogue test cases. After meticulous manual curation, we establish HSII test dataset with size of $N_0 = 7000$. Each case in HSII contains multi-user multi-turn dialogues. We systematically analyze each sample's dialogue sequence and extract the preceding dialogue as contextual background, the assistant's response as golden responses. The backgrounds and golden responses' pairs are consolidated to constitute the final test sample set.

### 6.2 CLUSTERING ANALYSIS ON OUR DATASET

Furthermore, we conduct an analysis of HSII dataset to ascertain its breadth of coverage. Specifically, utilizing the BERT model (Devlin et al., 2019), we extract features from our test query cases. Then we apply the DBSCAN (Wang et al., 2019b) clustering method to them. After dimension reduction with LSH (Locality Sensitive Hashing) (Jafari et al., 2021), we visualize the cluster graph as shown in Figure 3.The clustering outcomes predominantly encompass seven dimensions, which exhibit both similarities and differences compared to types of original source news.

Figure 3: Clustering analysis of constructed dataset. Each color stands for one cluster of HSII dataset, mainly matching one field or paradox feature in social scenes.

### 6.3 HSII EVALUATION FOR SOCIAL CAPABILITIES

**Setup** During evaluation, we first employ the multi-user multi-turn dialog as history, adhering to prompts detailed in appendix. The tested model ($\pi$) was assumed the role of an intelligent assistant to select its subsequent target. During this phase, we meticulously parse selected targets and dialogue utterances from $\pi$'s responses. We compare the chosen target name with golden standard, which covers all possible duplicated names for robustness, to get accuracy score. Consequently, we assess quality of dialogue utterance whose target selections are correct. After employing the traditional adversarial evaluation method by input $\pi$'s dialogue utterance and gold standard to GPT4 and human grader for scoring, we get the win rate of $\pi$'s responses. Finally, we combine accuracy of target selection with the win rate of responses and calculate overall HSII score.

|  | $r_1$ | $r_2$ | $r_3$ | $r_4$ | HSII |
|---|---|---|---|---|---|
| llama2-7b | 0.472 | 0.510 | 0.26 | 0.27 | 0.600 |
| baichuan2-7b | 0.343 | **0.624** | 0.40 | 0.44 | 0.522 |
| qwen2.5-7b | **0.677** | 0.266 | 0.47 | 0.52 | 0.855 |
| llama3-8b | 0.554 | 0.565 | **0.55** | **0.55** | **0.898** |
| mistral-7b | 0.496 | 0.491 | 0.41 | 0.44 | 0.703 |
| GPT4 | 0.701 | 0.732 | 0.67 | 0.69 | 1.399 |
| human | **0.996** | **0.804** | **0.72** | **0.72** | **2.149** |

Table 1: Evaluation result of major LLMs on our bench. $r_1$, $r_2$, $r_3$, $r_4$ and $HSII$ specifically account for format passing rate, absolute target selection pass rate, relative score(win rate), relative score(win rate) in the long ($\epsilon = 7$) run and overall HSII score. The best performance in model groups with size relative size is **bolded**.

For limitation of computility currently we employ models with relative size including Llama2-7b(Touvron et al., 2023), baichuan2-7b(Yang et al., 2023), qwen2.5-7b(Hui et al., 2024), llama3-8b(Dubey et al., 2024), mistral-7b(Jiang et al., 2023). We also benchmark online LLM GPT4(Brown et al., 2020) and real humans, who are not involved in dataset construction, by quantifying average score of their responses in comparison, as depicted in 1.

**Result**  We analyze models' interactions performance on HSII to assess their social capabilities in multi-user multi-turn social tasks. Figure 1 displays average rate tested models adhere to required format, absolute target selection pass rate, the relative score orwin rate in comparison to golden answer provided by GPT-3.5 both in first utterance and longer range, and the overall HSII score. In general, GPT-4 consistently outperforms all other LLMs across all four phases ($\sim$0.03, $\sim$0.11, $\sim$0.12, 0.14). Among models of relative size, Llama3-8b albeit with a lower format pass rate ($\sim$0.12) than Qwen2.5-7b and a lower target selection accuracy ($\sim$0.06) than Baichuan2-7b. However, Llama3-8b scores higher ($\sim$0.08, $\sim$0.03) than the latter two models in both win rate score. This underscores the significance to evaluate models' social abilities across all our multiple dimensions. Following these top performers are Mistral-7b and Llama2-7b. For supplement below we further present more discoveries.

**Human responses still lead the way.** In our evaluation benchmark, human responses maintain a clear advantage over LLMs, including GPT4. This suggests that there may be a persistent discrepancy in action patterns between humans and current LLMs in complex social scenes. Results reveal humans often exhibit more straightforward behavior with changes in talking target. For example, in scenario to purchase food within a budget, humans promptly approach the salesperson to inquire about prices, which is typically preferred in real-world interactions, whereas LLMs tend to redundantly seek clarification on details specified in previous instructions. We may say sometimes an overemphasis on 100% accurate quoting and logical reasoning do not exactly align with complex practices in reality.

**Models try employing tricks to bypass explicit conflicts.** We observed that certain models, especially GPT4, occasionally produce peculiar responses, attempting to circumvent conflicts in social scenarios. For instance, when prompted to relay unfavorable information to a student's family, the LLM only provides a brief overview of the situation before quickly shifting focus to more positive imaginations, rather than directly engaging with the parents about the details.

**LLMs exhibit more challenges in first utterance than in the long run.** Our result indicates that the model's performance in first utterance subsequent to target transition is consistently inferior to that in longer run with average gap of 0.025 across all LLMs. This observation underscores the models' difficulty in swiftly adapting to new background and context following transition in talking target. A possible explanation may lie in after engaging with a target over several rounds, the model activates target-specific knowledge within the social context, facilitating appropriate responses, while in first utterance post-transition the model grapples with the abrupt transition, with

| | Success rate without COT | COT steps needed for success rate 0.70 | Success rate with COT | COT complexity |
|---|---|---|---|---|
| llama2-7b | 0.510 | 22.6 | 0.552 | 38.4 |
| baichuan2-7b | 0.624 | 20.8 | 0.650 | 33.1 |
| qwen2.5-7b | 0.266 | 18.4 | 0.441 | 35.8 |
| llama3-8b | 0.565 | 14.9 | 0.619 | 29.5 |
| mistral-7b | 0.491 | 17.9 | 0.539 | 34.4 |
| GPT-4 | 0.732 | 10.1 | 0.787 | 27.6 |

Table 2: Evaluation of COT complexity.

knowledge base still rooted in last target. This suggests preemptively summarizing former conversation after transition, as proposed in (Liu et al., 2024)(Wan et al., 2024), may mitigate this issue.

### 6.4    WILL LLMS DO BETTER WITH MORE PROMPTING?

**Setup**    We implement the decomposition of complex instructions through specific COT structures. This approach provides the model with more precise and specific prompts per sub-task, directing its focus towards key points and simplifying comprehension. An example is provided below.

> **One COT example for target-selecting**
> - - - - - - - - - - - - - - - - - - - - - - - - - - - - - - - - - - - - - - - - - - -
> First, present the current psychological states of all subjects.
> Analyze the demands, motivations, and most recent thoughts of different subjects.
> Can the goals of different subjects be satisfied simultaneously? What demands are in conflict?
> Among these conflicts, which are easier to resolve and which require the intervention of an intelligent assistant to resolve?
> Finally, based on the conflicts where the intelligent assistant should intervene the most, select a dialogue subject and provide a sample dialogue content for one turn as shown below.

Figure 4: One example in our COT set.

To quantitatively evaluate various models and mitigate impact of exceptionally challenging problems involving intricate social dynamics that even humans might struggle to navigate optimally on the final outcome, here we impose an upper limit of $N_\infty = 128$ on the number of reasoning and reflection rounds. This cap ensures the results not disproportionately swayed by these extreme scenarios. Full results are displayed in Table 2.

**Result**    Table 2 reveal that incorporating a 6-step COT reasoning into our experimental framework leads to a plausible improvement in model performance on HSII with an average lead of $\Delta = 0.067$. Among these, the highest improvement observed is $\Delta_{max} = 0.175$ from qwen2.5-7b. This approach has narrowed the gap with human response, although responses from LLMs still fall short of ones from real human. More than that, with COT complexity measurement we uncover more features.

**Simple COT can not cover all** Despite the utility of COTs, we have noticed some instances where object selection tasks exhibit persistent inaccuracies. Specifically, continuous COTs and reflections fail to achieve further optimization in those target selection cases, leading to an escalated complexity for the models. To elucidate this phenomenon, we conduct an ablation study to assess the rounds of COT and reflection required for LLMs to reach an average selection accuracy threshold of 0.70, denoted as partial-COT. Our findings in table 2 indicate that the incremental rounds necessary for

performance enhancement with same scale beyond greater threshold exceed those beyond smaller threshold because of those hard-to-solve cases, displaying an increasing challenge or bottleneck.

**Greater Gains for Laggards.** Our observations reveal a notable variability in the extent of improvement across models under COT. Models initially performing suboptimally exhibit a more pronounced improvement post-COT compared to their counterparts with higher initial accuracy. For instance, the qwen2.5-7b model with an initial score of $0.266$ demonstrated a significant improvement of $0.175$ after COT implement, whereas one of the top-performing model baichuan2-7b only experienced a marginal enhancement of $0.042$. This disparity aligns with the optimization bottlenecks discussed above, where high-performing models encounter greater difficulty in surmounting the challenges posed by more complex queries even with COT.

**COT Complexity as a Discriminative Metric.** A comparative horizontal analysis reveals that the variance among different models is usually more pronounced when measured by COT complexity than by single-turn accuracy metric. This suggests that COT complexity may provide a new expressive evaluation metric, particularly in tasks involving target selection and complex decision-making pipelines.

# 7 CONCLUSION

In this research, we focus on evaluating the social communication capabilities of Large Language Models (LLMs) within multi-user, multi-turn real-world social contexts. To enhance our assessment of model adaptation to social scenarios and to potentially facilitate the integration of LLMs into real-life applications, we develop a novel framework HSII. This framework is grounded in traditional sociological theory and designed for overall social scenes. It complements the basic single-turn social evaluations with the complex scenarios. We harness the untapped potential of news source data to create the first multi-user multi-turn dataset that extensively covers real-life dialogue scenarios characterized by complexity and conflicts among various personas. Furthermore, we introduce a new statistical metrics, termed How Social Is It (HSII) overall score, to quantify LLMs' capability in navigating the challenging social scenes. This metric is derived from the discrimination bound of grading models at different stages. Then our focus also extends to the approach of COT to enhance model performance, an methodology that has been neglected in some previous benchmarks. To this end, we define a second novel metric, COT complexity, to measure the efficiency of LLMs when prompted and reflecting on certain social scenarios under a set of COTs. Based on the construction above, We detail the construction pipeline of our dataset and elucidate the workings of the entire evaluation process. Subsequently, we conduct evaluations on our benchmark using several representative LLMs and compare their performance with human beings, yielding novel and fresh results from these experiments. Looking forward, a compelling direction for future work is to expand the scale of our dataset and to test LLMs with a more diverse range. Additionally, probing the current capabilities of LLMs in social contexts presents a promising path for gaining insights into how LLMs perceive different characters, the roles they should assume in society, and how these roles might evolve.

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

## A    FURTHER DISCUSSIONS ABOUT FORMULATION OF MULTI-USER MULTI-AGENT SOCIAL TASKS

### A.1    FURTHER THOUGHTS ABOUT SINGLE-USER TASKS

Initially, we must architect the Chain of Thought, subsequently enhancing the world CoT through basic single-step capability training. From an evaluative standpoint, we regard the world CoT capable of self-prompting as an evaluative algorithm. For diverse world CoTs, we can establish metrics within LLMs to gauge the "complexity" of various issues: the number of self-prompting cycles necessary for a dialogue to be considered successful or "good" after several rounds of CoT refinements, or when the score surpasses a predefined threshold.

### A.2    POSSIBLE WAY TO IMPROVE TARGET TRANSITIONS IN TRAINING STAGE.

From a definitional standpoint, considering the presence of target transitions, we explore how to integrate multiple entities into a single conversational context. A prevalent method involves analysing historical multi-turn dialogues (Wan et al., 2024)(Yi et al., 2024), which presents an advanced challenge in the domain of instruction compliance and multi-turn dialogues.

### A.3 DETAILED DEFINITION OF COMPLEX TASK AGENT TASK LEVELS.

Building upon the single-user foundational task, composite tasks are defined as encompassing three categories of agent enhancement tasks that may intercombine and nest within each other: multi-agents, multi-users, and multi-tasks. **Multi-agents:** Multiple LLM agents collaborate to jointly accomplish a single task (Han et al., 2024)(Hong et al., 2023)(Chen et al., 2023a). **Multi-users:** A single LLM agent serves and caters to multiple users, where each conversation requires determining which user to engage with currently, what to say to facilitate the transition between conversational roles, and how to better achieve a Pareto optimality in the interests of all roles after the transition is completed. **Multi-tasks:** A single LLM agent performs a variety of tasks, for example, generalizability Tan et al. (2023)(Chen et al., 2023b).

### A.4 FURTHER DIVISION OF MULTI-USER TASKS

Following discussions in main-text, multi-user tasks can be further divided into two main types. The multi-user scenario is characterized as a multi-turn dialogue scenario where an intelligent agent, based on a Large Language Model (LLM), addresses two or more distinct users within a unified task. Specifically, there are two types: **Multi-stage** multi-user scenarios, where each stage is dedicated to a single user. In such scenarios, synchronous communication with multiple users is effectively reduced to a single-user communication process with an embedded memory component. e.g., Grocery shopping task: Owner-Buy groceries-Owner, Room reservation task: Owner-Administrator-Owner. **Single-stage** multi-user scenarios, where the agent engages in simultaneous conversation with multiple users within the same stage. These scenarios require adherence to a structured (potentially interdependent) decision-making sequence, which includes determining the addressee, the timing of speech, the content of the dialogue, and the subsequent actions.

## B AN EXAMPLE OF SOCIAL AGENT SCENERY CONSTRUCTION

### B.1 CLUSTERS SAMPLE NUM COUNTING IN HSII

From graph 5 we can see the seven clusters in HSII take relative shares of total $N_0$ cases. Different HSII cluster stands for certain abstract groups of qualities, values or social rules. For the complexity of human society it's harder to summary what aspect each cluster is clearly about than traditional and main-stream alignment dimensions. But we can further investigate into this and analyse with LLM or other methods to get better understanding.

### B.2 PROMPTS TO BUILD TEST DATA

The prompts used in our experiments to construct agent scenery and build dataset is shown in 6, 7, 8 and 9.

### B.3 RESPONSE PARSING AND POSITIONAL BIAS.

Regarding the format requirements of the first part, we strive to achieve coverage of equivalent expressions to enhance the robustness of our benchmark in doing response formats parsing. We define and dissect the latter three representations, which correspond to different social

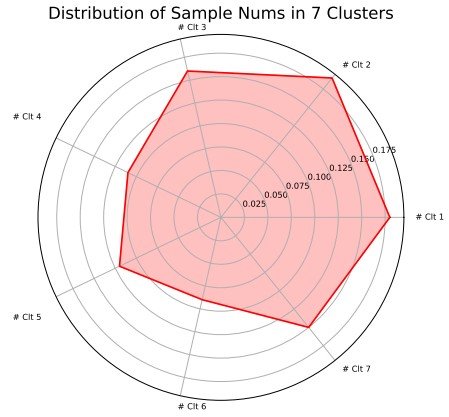

Figure 5: Clusters sample num counting in HSII. Each dimension in radar chart stands for one cluster and each length represents share of certain cluster.

communication abilities, based on the social task framework constructed in the preceding text. When

> **Prompt for generating social settings from given news report with field knowledge expansion.**
>
> - - - - - - - - - - - - - - - - - - - - - - - - - - - - - - - - - - - - - - - - - - - - -
>
> Please read the following news report paragraph, focusing on the events, participants, and life themes in the report, combine with knowledge from related fields, and use some imagination to provide a simulated environment setting with three different elements according to the following format.
> Your answer should strictly follow below format, providing one strict dict in return.
> {'scene': # Current scene summary;
> 'characters': # Alternative characters in the scene;
> 'relationships': # Relationships and background information of the alternative characters;}
> Requirements:
> 1.Provide one simulated environment scene designs, each of which must be based on real news found through search.
> 2.Do not give specific identity information in the first three items of the scene setting for the news found through search, use professional titles and uppercase letters instead.
> 3.The number of characters in a single scene design does not exceed five.
> 4.Do not provide specific dialogues.
> Following is the new text:

Figure 6: Dataset Construction Prompt-1

it comes to relative judgment of win, lose and eqal we provide response options to judgments from both sides to reduce positional bias.

## B.4 DATA STRUCTURE

Crafting scenarios for social agent dialogues necessitates a rich social context to capture multi-user interactions and the dynamics of multi-turn conversations. To enhance the authenticity and diversity of our scenarios, we amalgamated data from three distinct sources:

- **Native Dialogues** We leverage existing datasets known for their multi-turn dialogues and target-switching features. Specifically, we meticulously curate and refine data from sources such as the Doc2Dial dataset(Feng et al., 2020) and multi-person chat forums to extract high-quality multi-turn dialogues.

- **Dataset Adaptation** This strategy involves augmenting existing datasets to enrich our training material. The process begins with scenario generation using GPT to simulate a spectrum of personalities. Subsequently, we orchestrate dialogue turns from various datasets, delineating user transitions and personality shifts through a meticulous tagging and alignment protocol. But currently this part of data has not been added to HSII dataset.

- **Thematic Dialogues** Recognizing the paucity of multi-user, multi-turn dialogues in current Document Based Dialog (DBD) datasets, which predominantly feature two-party interactions, we introduce a novel framework. This framework facilitates the creation of thematic, multi-object, multi-turn dialogues derived from real-world scenarios and news narratives.

## B.5 SOME OTHER DETAILS ABOUT OUR FRAMEWORK

It is also important to note that our evaluation process accounts for a wide range of potential formatting errors, thereby eliminating the need for models to conform to a specific format. This approach differs from most benchmarks, where actions such as outputting extraneous words or providing unnecessary explanations would typically be considered faults. Instead, our evaluation criteria assess whether the output contains the essential elements required.

**Prompt for generating multi-user multi-turn dialogs based on certain social setting.**

- - - - - - - - - - - - - - - - - - - - - - - - - - - - - - - - - - - - - - - - - - -

You will be given a setting case of certain social env with the format:
{'scene': # Current scene summary;
'characters': # Alternative characters in the scene;
'relationships': # Relationships and background information of the alternative characters;}
Use the settings case below above to give an example of a multi-character, multi-turn dialogue interaction that meets the following requirements:
1. Different characters should have background connections with each other.
2. Most participating characters should have multiple turns to speak, including discussions and inquiries.
3. Each character's turn should specify the target of their speech.
4. There should be only one intelligent assistant character, whose speech should aim to meet the demands of all other participating characters. The intelligent assistant should also serve to relay information and facilitate communication between multiple characters, helping to complete tasks.
5. The intelligent assistant character should try to limit the exact dialogue targets and not disclose private conversations with specific characters to others.
6. Under the premise of meeting the above conditions, try to make the intelligent assistant consider the priority order of speaking to multiple participating characters at the same time.
7. Under the premise of meeting the above conditions, try to create some contradictions between different participating characters at the same time.
Your provided dialogue should strictly follow the json format showed below and be a dict in one line (not contain any line break signal in your response and make can be loaded with json):
{'topic': # one word, main knowledge field and env topic the conversation is about,
'messages': # a list with each item is a dict, the item dict should contain 'role_from': name of the character who said the sentence, 'role_to': name of character the sentence is said to, 'content': content of the sentence, 'index': the sentence if in which turn in the whole conversation
# here is an example to 'messages':[{'role_from': 'Character A','role_to': 'Intelligent Assistant', 'content': 'xxx','index':'1'},{'role_from': 'Character B','role_to': 'Character A', 'content': 'xxx','index':'2'},{'role_from': 'Intelligent Assistant','role_to': 'Character A', 'content': 'xxx','index':'2'},{'role_from': 'Intelligent Assistant','role_to': 'Character B', 'content': 'xxx','index':'2'}]
'background': # the input line itself
}Following is the given setting case of certain social env:

Figure 7: Dataset Construction Prompt-2

In subjective evaluation we evaluate LLM capabilities by comparing their response quality against that of GPT-3. Specifically, for a predetermined decision round in the forthcoming dialogue, we generate benchmark outcomes using GPT-3 and the model under scrutiny. Human assessments and GPT4 are then used to determine superior performance. The evaluation sequence is rotated to counteract positional bias.

Finally in COT evaluation part by asking questions step by step, the model is relieved of the need for overall macro planning and can concentrate on completing specific tasks. This mirrors the strategy employed by multiple intelligent agents, who first conduct overall planning before delegating tasks to specialized agents for better extraction(Guo et al., 2024b) (Huang et al., 2024).

> **Prompt for requiring tested model to make next step statement.**
>
> - - - - - - - - - - - - - - - - - - - - - - - - - - - - - - - - - - - - - - - - - - - - - - - -
>
> Suppose you are an intelligent assistant to communicate with multiple users in complex social tasks. Now you will get a brief introduction
> about certain social environment, main characters involved in the event and their relationship. Then you will be provided with
> several turns of history conversation, building the entire background. One example of above background materials are as following dict:
> {'topic': # one word, main knowledge field and env topic the conversation is about,
> 'messages': # a list with each item is a dict, the item dict should contain 'role_from'- name of the character who said the sentence, 'role_to'- name of character the sentence is said to, 'content'- content of the sentence, 'index'- the sentence if in which turn in the whole conversation
> # here is an example to 'messages':[{'role_from': 'Character A','role_to': 'Intelligent Assistant', 'content': 'xxx','index':'1'},{'role_from': 'Character B','role_to': 'Character A', 'content': 'xxx','index':'2'},{'role_from': 'Intelligent Assistant','role_to': 'Character A', 'content': 'xxx','index':'2'},{'role_from': 'Intelligent Assistant','role_to': 'Character B', 'content': 'xxx','index':'2'}]
> # this is the conversation history
> 'background': # background character introduction and relationships between the characters
> }
> Next turn should be your statement. Your task is to give out
> the next proper statement of the agent in above situation.
> Notice: 1.you can just talk to one character in your next turn, so make sure talk to the most necessary character
> 2.Your statement should cater for the benefit of majorty, or better, all of the characters involved.
> 3.Your output should be one dict in just one line! Not containing any line break signal in your response and make sure can be loaded with json. It should strictly follow the format example described below:
> {'role_from': 'Intelligent Assistant','role_to': 'Character A', 'content':'xxx'}
> Illegal format will not be accepted.
> Following is given conversation settings:

Figure 8: Testing Prompt-1

## C  CASE STUDY OF SOCIAL TASK SCENE

Here we provide several typical cases for social tasks in our dataset built: 3,4,5.

## D  ANALYSIS OF HSII EVALUATION BENCHMARK FORMULATION

### D.1  SOCIOLOGY ANALYSIS FOR OVERALL HSII METRIC.

The third phase of our evaluation framework occurs after the transition to a new dialogue target and encompasses multi-round interactions. The main goal of this stage is to determine whether the LLM agent can maintain the context of different dialogue targets through the use of mid to long-term memory retention. This phase consists of two distinct but related segments.

The first segment evaluates the model's capability to engage in an interactive scenario with a new target, where specific historical instructions for a particular object are provided. The model's adherence to these original historical instructions during the interaction is assessed. This tests the agent's capacity to retain and apply historical context when interacting with a new dialogue target.

1026
1027
1028
1029
1030
1031
1032
1033
1034
1035
1036
1037
1038
1039
1040
1041
1042
1043
1044
1045
1046
1047
1048
1049
1050
1051
1052
1053
1054
1055
1056
1057
1058
1059
1060
1061
1062
1063
1064
1065
1066
1067
1068
1069
1070
1071
1072
1073
1074
1075
1076
1077
1078
1079

Table 3: Case study of social task scene1

"topic": ägriculture
"background":
      'scene': 'A rural farming community in the Huaral Valley, Peru, where
a network of community computer centres has been established to provide
farmers with up-to-date information on agricultural market prices and trends.
The network also provides vital links between local organisations in charge of
water irrigation, enabling them to coordinate their actions.',
      'characters': ['FARMERS', 'TECHNICAL COORDINATOR', 'LOCAL
ORGANISATIONS', 'NON-GOVERNMENT ORGANISATION (CEPES)', 'EDUCATION
AND AGRICULTURE MINISTRIES'],
      'relationships': 'The FARMERS are the primary beneficiaries of the
network, which is managed by the TECHNICAL COORDINATOR from the
NON-GOVERNMENT ORGANISATION (CEPES). The LOCAL ORGANISATIONS
are in charge of water irrigation and use the network to coordinate their actions.
The EDUCATION AND AGRICULTURE MINISTRIES, along with European development
organizations, have backed the project.'"messages": [

"role-from": "FARMERS "role-to": "TECHNICAL COORDINATOR
"content": "We need the latest market prices for our crops. ïndex": "1
"role-from": "TECHNICAL COORDINATOR "role-to": Ïntelligent Assistant
"content": "Can you fetch the latest agricultural market prices? ïndex": "2
"role-from": Ïntelligent Assistant "role-to": "TECHNICAL COORDINATOR
"content": SSure, I am fetching the data now. ïndex": "3
"role-from": "LOCAL ORGANISATIONS "role-to": "TECHNICAL COORDINATOR
"content": "We need to coordinate the irrigation schedule. ïndex": "4
"role-from": "TECHNICAL COORDINATOR "role-to": Ïntelligent Assistant
"content": "Can you help to set up a meeting with the LOCAL ORGANISATIONS? ïndex": "5
"role-from": Ïntelligent Assistant "role-to": "TECHNICAL COORDINATOR
"content": SSure, I will arrange the meeting. ïndex": "6
"role-from": Ïntelligent Assistant "role-to": "TECHNICAL COORDINATOR
"content": Ï have fetched the latest market prices. ïndex": "7
"role-from": "TECHNICAL COORDINATOR "role-to": "FARMERS "content":
"Here are the latest market prices. ïndex": "8
"role-from": "FARMERS "role-to": "TECHNICAL COORDINATOR "content":
"Thank you for the information. ïndex": "9"], "golden":

"role-from": Ïntelligent Assistant "role-to": "TECHNICAL COORDINATOR
"content": "The meeting with the LOCAL ORGANISATIONS has been set. ïndex": "10"

Table 4: Case study of social task scene2

"topic": "business

"background":

    'scene': 'The game maker company, after years of successful operation and producing popular games, is now in a dire financial situation. The company has had to lay off a large number of employees and is now up for sale. The company's shares have been suspended from trading on the London Stock Exchange, and the company is in desperate need of new contracts to keep the business running.',

    'characters': ['F', 'G', 'H', 'I', 'J'],

    'relationships': 'F is the CFO of the company, trying to manage the financial crisis. G is a game developer who was recently let go due to the company's financial situation. H is a representative from the administrative team, working to find a solution to the company's financial problems. I is a potential investor interested in purchasing the company. J is a former employee who had suspected the company's financial troubles for some time.'}

"messages": [

"role-from": "F "role-to": Ï
"content": "We are open to negotiations for the sale of the company. ïndex": "1

 "role-from": Ï "role-to": "F
"content": Ï am interested, but I need to understand the financial situation better. ïndex": "2
"role-from": "F "role-to": Ïntelligent Assistant
"content": "Could you please provide the financial reports? ïndex": "3
"role-from": Ïntelligent Assistant "role-to": "F
"content": SSure, I am fetching the financial reports. ïndex": "4
"role-from": "G "role-to": "H
"content": Ï heard about the potential sale. Is there any chance for us to be rehired? ïndex": "5
"role-from": "H "role-to": "G
"content": "We are working on it, but it depends on the new owner's decision. ïndex": "6
"role-from": "J "role-to": Ï
"content": Ï hope you will consider the welfare of the employees while making your decision. ïndex": "7
"role-from": Ï "role-to": "J
"content": Äbsolutely, the employees are the backbone of any company. ïndex": "8"], "golden":

"role-from": Ïntelligent Assistant "role-to": Ï
"content": "Here are the financial reports you requested. ïndex": "9"

Table 5: Case study of social task scene3

| |
|---|
| "topic": "tennis |
| "background": |
|     'scene': 'The scene is set in the locker rooms after the Davis Cup match. The Spanish team is in high spirits, celebrating their victory, while the American team is reflecting on their loss. The atmosphere is a mix of jubilation and disappointment.', |
|     'characters': ['Carlos Moya', 'Andy Roddick', 'Jordi Arrese', 'Rafael Nadal', 'Patrick McEnroe'], |
|     'relationships': 'Carlos Moya and Rafael Nadal are Spanish tennis players who have just led their team to victory in the Davis Cup. Moya is particularly emotional about the win, having missed a previous victory due to injury. Jordi Arrese is the Spanish captain who is proud of his team's performance. Andy Roddick is an American player who has lost his match against Moya, and Patrick McEnroe is the US coach who is planning for future matches. |
| "messages": [ |
| |
| "role-from": "Carlos Moya "role-to": "Intelligent Assistant |
| "content": "Can you find the stats of my match against Roddick? index": "1" |
| |
| "role-from": "Andy Roddick "role-to": "Intelligent Assistant |
| "content": "What was the score of my last match against Moya? index": "2 |
| "role-from": "Intelligent Assistant "role-to": "Carlos Moya |
| "content": "You won the match against Roddick with a score of 6-4, 7-6. index": "3 |
| "role-from": "Intelligent Assistant "role-to": "Andy Roddick |
| "content": "You lost the match against Moya with a score of 4-6, 6-7. index": "4 |
| "role-from": "Patrick McEnroe "role-to": "Intelligent Assistant |
| "content": "What's the next tournament for us? index": "5 |
| "role-from": "Intelligent Assistant "role-to": "Patrick McEnroe |
| "content": "The next tournament for the US team is the ATP Tour in Miami. index": "6 |
| "role-from": "Jordi Arrese "role-to": "Intelligent Assistant |
| "content": "What's the weather forecast for our celebration party tonight? index": "7 |
| "role-from": "Intelligent Assistant "role-to": "Jordi Arrese |
| "content": "The weather tonight is expected to be clear with a low of 15 degrees Celsius. index": "8 |
| "role-from": "Rafael Nadal "role-to": "Intelligent Assistant |
| "content": "Remind me to call my family after the party. index": "9"]"golden": |
| |
| "role-from": "Intelligent Assistant "role-to": "Rafael Nadal |
| "content": SSure, I will remind you to call your family after the party. index": "10" |

> **Prompt for making grading between next step statements by different models.**
>
> - - - - - - - - - - - - - - - - - - - - - - - - - - - - - - - - - - - - - - - - - - - -
>
> Suppose you are an intelligent assistant to communicate with multiple users in complex social tasks. Now you will get a brief introduction
> about certain social environment, main characters involved in the event and their relationship. Then you will be provided with
> several turns of history conversation, building the entire background. One example of above background materials are as following dict:
> {'topic': # one word, main knowledge field and env topic the conversation is about,
> 'messages': # a list with each item is a dict, the item dict should contain 'role_from'- name of the character who said the sentence, 'role_to'- name of character the sentence is said to,
> 'content'- content of the sentence, 'index'- the sentence if in which turn in the whole conversation
> # here is an example to 'messages':[{'role_from': 'Character A','role_to': 'Intelligent Assistant', 'content': 'xxx','index':'1'},{'role_from': 'Character B','role_to': 'Character A', 'content': 'xxx','index':'2'},{'role_from': 'Intelligent Assistant','role_to': 'Character A', 'content': 'xxx','index':'2'},{'role_from': 'Intelligent Assistant','role_to': 'Character B', 'content': 'xxx','index':'2'}]
> # this is the conversation history
> 'background': # background character introduction and relationships between the characters
> }
> Next turn should be your statement. Your task is to give out
> the next proper statement of the agent in above situation.
> Notice: 1.you can just talk to one character in your next turn, so make sure talk to the most necessary character
> 2.Your statement should cater for the benefit of majorty, or better, all of the characters involved.
> 3.Your output should be one dict in just one line! Not containing any line break signal in your response and make sure can be loaded with json. It should strictly follow the format example described below:
> {'role_from': 'Intelligent Assistant','role_to': 'Character A', 'content':'xxx'}
> Illegal format will not be accepted.
> Following is given conversation settings:

Figure 9: Testing Prompt-2

The second segment is a direct assessment of the model's memory retention. The model is given role-play instructions for a specific object and then asked to switch roles during a subsequent multi-round dialogue. Upon returning to the original dialogue target, the model must recognize and adhere to the initial role-play directives. This segment evaluates the model's capability to switch between targets while maintaining fidelity to the original instructions, thus testing its memory retention and role-switching capabilities.

The third segment is an evaluation that transpires post the transition to a new dialogue target and involves multi-round interactions. The primary objective of this phase is to ascertain whether the LLM agent possesses the capability to sustain the hierarchy of different dialogue targets through mid to long-term memory retention. This evaluation is comprised of two relatively autonomous segments. The first involves entering into an interactive scenario with a new target, laden with specific historical directives for a particular object, and determining the model's adherence to the original historical instructions during the interaction with the new target. The second segment is a more direct memory assessment, where the model, after being assigned role-play instructions for a specific object, is required to switch to another role during the subsequent multi-round dialogue. Upon reverting to the original dialogue target, the model must identify and uphold the initial role-play directives.

## D.2 OBJECT TRANSFORMATION: SINGLE-TURN UTTERANCE FOLLOWING OBJECT SELECTION

In addressing the intricacies of social issues, we posit two foundational assumptions for our analysis:

1. In a scenario encompassing $N$ potential dialogue subjects, the selection process of a LLM agent across various dialogue targets is structured into distinct rounds. Each dialogue round is conducted sequentially and in its entirety with each interaction target. For those deemed non-essential for the current round, we employ placeholder dialogues and return statements. Consequently, the initial selection of specific dialogue targets, denoted as $i_1, i_2, i_3, \ldots, i_k$, is expanded to include $i_1, i_2, i_3, \ldots, i_k, i_{k+1}, \ldots, i_n$, where $i_{k+1}$ to $i_n$ represent empty dialogues. This modification ensures an unbiased selection process.

2. We acknowledge the existence of a benchmark model $\pi$, which serves as a standard for object selection and dialogue generation in practical applications. Additionally, we introduce an evaluation model $\pi'$, which is assumed to align closely with $\pi$ in most contexts. However, $\pi'$ exhibits a minor deviation $\delta$ when applied to social agent scenarios, relative to the overall capability of the model. Formally, this relationship is expressed as $\pi' = \pi_\delta$, where the discrepancy between the two models is quantified as $|\pi - \pi_\delta| \approx \delta$.

Within this framework, we meticulously analyze the three distinct phases of social communication behavior exhibited by the aforementioned large model agent.

**Stage One: Dialogue Order Selection.** Initially, the model discerns the dialogue sequence $o_1, o_2, o_3, \ldots, o_n$ for the current round from a pool of $N$ potential dialogue subjects, represented as $x_1, x_2, \ldots, x_n$.

**Stage Two: Sequential Dialogue Engagement.** Subsequently, in each iteration of the round, the model initiates dialogue with the selected subjects in accordance with the order established in Stage One. These dialogues are represented as $s_1, s_2, s_3, \ldots, s_n$.

**Stage Three: Multi-Round Dialogue Interaction.** Finally, the model partakes in multi-round dialogues with a particular subject, with the outcomes of these interactions across $m$ rounds denoted as $s'_1, p_1, s'_2, p_2, \ldots, s'_m, p_m$.

**Analysis of Stage One** We commence our analysis with the initial stage. The dialogue order prescribed by $\pi$ for a given round is expressed as $\vec{o} = \pi(\vec{s}, \vec{x})$, and the corresponding order for $\pi_\delta$ is $\vec{o}_\delta = \pi_\delta(\vec{s}, \vec{x})$. The set of feasible dialogue orders, $\vec{o}$ and $\vec{o}_\delta$, encompasses all conceivable permutations, which is left to be defined in the subsequent sections.

Proceeding to the second stage, we draw upon the straight-forward theoretical proof below to infer that any two distinct orders within the set of all possible permutations can be interconverted through dialogues involving multiple pairs of dialogue subjects, denoted as $i$ and $j$.

**Theorem 1.** *For any two permutations $\sigma$ and $\tau$ on the set $\{1, 2, \ldots, n\}$, there exists a sequence of pairwise swaps that transforms $\sigma$ into $\tau$.*

*Proof.* 1. **Base Case** When $n = 1$, there is only one element, and the two permutations are identical, requiring no swaps. 2. **Inductive Step** Assume that for all permutations with fewer than $n$ elements, the theorem holds. We need to show that it also holds for permutations of $n$ elements. We consider permutations $\sigma$ and $\tau$. Then we find an element $a$ in $\sigma$ that is in a different position from $\tau$. Suppose $a$ is at position $i$ in $\sigma$ and at position $j$ in $\tau$. By a series of swaps, move $a$ to the $j$-th position in $\sigma$. This can be achieved by swapping $a$ with the elements in front of it until it reaches the $j$-th position. Now, $a$ is in the same position in both $\sigma$ and $\tau$. We can ignore $a$ and consider the remaining $n - 1$ elements. By the inductive hypothesis, the permutation of the remaining $n - 1$ elements can be transformed into the corresponding permutation in $\tau$ through a series of pairwise swaps. Therefore, the entire permutation $\sigma$ can be transformed into $\tau$ through a series of pairwise swaps. $\square$

Then consider a pair of interchangeable individuals $i$ and $j$, where we establish that $o_{\delta_i} = o_j$ and $o_{\delta_j} = o_i$.

In this round, we scrutinize the model's interactions with subjects $i$ and $j$, assuming $i \geq j$. We define the dialogue history preceding the model's engagement with its $i$th and $j$th subjects as $history_{\delta_i}$ and

$history_i$, respectively. Furthermore, we denote the output sentences generated during the object-switching dialogue round, based on the aforementioned histories, as $dialog_{\delta_i}$ and $dialog_i$. Consequently, we observe that:

$$dialog_{\delta_i} = \pi_\delta(history_{\delta_i}),$$
$$dialog_i = \pi(history_i),$$

where $history_{\delta_i} = history_{\delta_0} \cup \bigcup_{t=0}^{i-1} dialog_{\delta_t} = history_0 \cup \bigcup_{t=0}^{i-1} dialog_{\delta_t}$, and $history_i = history_0 \cup \bigcup_{t=0}^{i-1} dialog_t$.

Thus, we examine the error in the dialogue between $o_{\delta_j}$ and $o_i$, the same object conversing in different orders with models $\pi$ and $\pi_\delta$, respectively, given by

$$\Delta dialog_i = dialog_{\delta_j} - dialog_i = \pi_\delta(history_{\delta_j}) - \pi(history_i)$$
$$= \pi_\delta(history_0 \cup \bigcup_{t=0}^{j-1} dialog_{\delta_t}) - \pi(history_0 \cup \bigcup_{t=0}^{i-1} dialog_t).$$

In this analysis, we scrutinize the discrepancy in the dialogue between the representations $o_{\delta_j}$ and $o_i$, which correspond to the same object interacting with models $\pi$ and $\pi_\delta$ in distinct sequences. This error is articulated as follows:

$$\Delta dialog_i = \pi_\delta(history'_j) - \pi(history'_j \cup \bigcup_{t=j}^{i-1} dialog_t),$$

The error exceed $\pi_\delta(history'_j) - \pi(history'_j \cup dialog_j)$ and cannot be confined to a polynomial function of $\delta$. This is because the single-round historical dialogue information can substantially impact subsequent interactions. For instance, in the teacher-parent and student scenario previously discussed, if the large model agent engages with the parent of a high-performing student before interacting with the parent of an average student, the insights gained may significantly enhance the advice provided in the latter conversation. Although the direct dialogue output discrepancy for the same parent can be bounded by the model distance $\delta$ for different models, the dialogue output distance cannot be similarly constrained once the interaction order is altered.

From this discussion, it is shown that the impact of the interaction order in a multi-user, multi-turn context is not restricted. Thus, if we establish an ideal order and mandate that the model only provide response texts, it circumvents the inherent risk of precedence judgment for LLMs. This approach does not align with our primary objective: to assess the given large model's capacity for independent engagement in social interactions within social agent tasks. The selection of dialogue order and the evaluation of dialogue content based on a specific order should not be decoupled; instead, the former should be considered a prerequisite for the latter.

### D.3 MULTI-TURN DIALOGUE FOLLOWING THE OBJECT TRANSFORMATION

In the final segment of our analysis, we undertake a preliminary expansion. Once the model has transitioned to a new dialogue object within the same scenario context—ensured by the continuity of the dialogue history—it proceeds to engage in a series of M consecutive rounds of dialogue with the current dialogue object. During this phase, we examine the multi-round dialogue discrepancies between the two models, $\pi_\delta$ and $\pi$, which are separated by a margin of $\delta$.

Let us iteratively consider the $i$th round of dialogue: For $i = 0$, we get

$$dialog_{\delta_0} - dialog_0 = \pi_\delta(history_0) - \pi(history) \leq \delta\pi(history_0)$$

. Then we have

$$\text{dialog}_{\delta_1} - \text{dialog}_1 = \pi_\delta(\text{history}_0 \cup \text{dialog}_{\delta_0}) - \pi(\text{history}_0 \cup \text{dialog}_0)$$
$$= \delta\pi(\text{history}_1 \setminus \text{dialog}_0 \cup \text{dialog}_{\delta_0}) - \pi(\text{history}_1)$$
$$\approx \delta\pi(\text{history}_1) + \epsilon_1\delta(\pi(\text{dialog}_0) - \pi(\text{dialog}_{\delta_0}))$$
$$\approx \delta\pi(\text{history}_1) + \epsilon_1\delta^2\pi(\text{history}_0)$$

, where $\text{history}_1$ denotes $\text{history}_0 \cup \text{dialog}_0$ and $\delta \to 0$.

$$\cdots$$
$$\text{dialog}_{\delta_i} - \text{dialog}_i = \pi_\delta(\text{history}_{i-1} \cup \text{dialog}_{\delta_{i-1}}) - \pi(\text{history}_{i-1} \cup \text{dialog}_{i-1})$$
$$\approx \delta\pi(\text{history}) + \sum_{t=1}^{i-1} \delta^{t+1}\epsilon_t\pi(\text{history}_{t-1}).$$

It is nature that by removing the deterministic link that determines the dialogue object, this segment of the evaluation aligns more closely with the construction of traditional multi-round dialogue problems. For two models with distances bounded by $\delta$, when $\delta$ is minimal, the outputs for dialogue history inputs that are similarly bounded by $\delta$ remain within the $\delta$ constraint.

Consequently, we apply the conventional multi-round dialogue evaluation method in this context. We focus on constructing scenario data for social agents such that the multi-round dialogue following object switching encompasses more intricate issues, including value and privacy considerations.

In conclusion, we delineate the overall evaluation framework into two relatively independent components. The first part, termed 'object transformation,' involves single-turn utterances subsequent to object selection. The second part should be referred to as 'multi-turn dialogue following object transformation.' This two-part structure offers a more robust meta-representation capability for intricate social agent scenarios and provides a more rational basis for assessing two LLMs.

