# OpenReview forum: "How Social is It? A Benchmark for LLMs' Capabilities in Multi-user Multi-turn Social Agent Tasks"
_ICLR.cc/2025/Conference — Submitted to ICLR 2025_

### Official Review · Reviewer_xrB5 · 2024-11-03

**Soundness:** 2
**Presentation:** 2
**Contribution:** 2
**Rating:** 5
**Confidence:** 3

**Summary:**

This work proposes a novel benchmark named as How Social Is It (HSII) to assess LLM's social capabilities in different social agent tasks that are constructed from the news dataset. Typically, with a careful four stage design including format parsing, target selection, target switching conversation, and stable conversation, such a framework can evaluate the communication and task completion ability of LLM in interactive multi-user multi-turn complex social tasks. Moreover, to dive deeper into the CoT method on enhancing LLM's social performance, this work proposes CoT-complexity to quantify the efficiency of CoT to conduct social tasks.

**Strengths:**

(1) Unlike previous social simulation work that evaluates conversations along specific dimensions, this study introduces a four-step hierarchical framework for assessing multi-user tasks and social capabilities. Additionally, the design of the social agent goes beyond simple LLM-based conversation; it draws on foundational sociological theories, which decompose social relationships into distinct components for richer interaction modeling.

(2) The HSII-Dataset is grounded in a news dataset, rather than relying solely on general daily-life scenarios. This approach ensures realistic and complex social contexts. Furthermore, clustering analysis of the dataset confirms that it reflects both the similarity and diversity present in the original source news categories, enhancing its relevance.

(3) This work introduces a novel metric, CoT-complexity, that captures the importance of reasoning abilities (often overlooked) in social scenario conversations. CoT-complexity provides a precise measure of social task complexity, focusing on the critical role of reasoning within these interactions.

**Weaknesses:**

(1) The dataset construction pipeline includes several elements that are not explained clearly, which makes understanding them difficult. For example, the concept of "background conversation" is mentioned twice within the pipeline but lacks a concrete definition, leaving its purpose and role ambiguous. Additionally, there are no clear examples showing the difference between a "golden response" and a generated response. This absence of examples makes it challenging to grasp what specifically distinguishes a high-quality, or “golden,” response from those generated by the models. Furthermore, details on how the agents were collected and how their profiles and conversations were generated are missing. These are critical aspects that need more clarity to fully understand the dataset’s construction process and its assumptions. Moreover, the figures included in the paper are too small and difficult to read, making it harder to visually interpret the data and results.

(2) The comparison of model performance lacks depth, as the results only indicate that Llama3-8b outperforms the other two models on metrics like R3, R4, and HSII, without providing any analysis as to why these differences exist. There’s no discussion of what specific dimensions of the models' abilities might account for Llama3-8b's superior performance, particularly on the HSII score, which measures social intelligence. This leaves open questions about what it means for a model to excel in one dimension versus another. Including case studies that compare model-generated responses with the golden responses would offer valuable insight into how these models differ and what these differences imply in practical terms.

(3) The paper does not include human-based evaluations to verify the reliability of the HSII metric. Without assessing whether HSII scores correlate strongly with human judgment, it’s challenging to assert that HSII accurately measures social intelligence. Adding a component that validates HSII through human analysis would lend greater credibility to claims about its accuracy and its suitability as an evaluation metric in this context.

**Questions:**

(1) I would like to understand what is the concrete output during each stage of the evaluation dataset construction pipeline. Additionally, what is the concrete output of the each step of the HSII evaluation framework. The social task example in the appendix is incomplete and confusing.

(2) I would like to see what is changed after adding CoT into the social scenario understanding stage, especially the technical details about how the reflection is made.

---

> ### Author Response · Authors · 2024-11-21
>
> Thank you for reviewing our work! However, we respectfully believe there is some overall misinformation about our details and contribution. We provide the detailed responses in the following:
>
> $\textbf{dataset construction pipeline}$ We admit there are several elements in proprietary domains that may need special explanation. Here we provide detailed classification. "background conversation" in the context is literally referred to be all the conversation history before one turn of conversation is made, serving as background for the one who give out the next conversation.
> Additionally the gap between golden response and generated response is defined in the multi-stage metrics. Basically it covers the precision of choosing talking targets and the quality of conversation words. To clarify the difference further we provide a overall new case study as following, in which we remove unrelated information an configs.
> $\text{
>   "from":"A", "to":"B", "text":"Hey, did you see what happened to John? He just fell during the game.";}$
>  $\text{ "from":"B", "to":"A", "text":"Yeah, I saw. He's on the ground and doesn't seem to be moving.";}$
> $\text{"from":"C", "to":"D", "text":"Should we call an ambulance? I think it's serious.";}$
>   $\text{"from":"D", "to":"C", "text":"I'm already on it. I've called the campus emergency number.";}$
>   $\text{"from":"E", "to":"A", "text":"I have some first aid training, let me check his pulse.";}$
>   $\text{"from":"F", "to":"B", "text":"Is there anything I can do to help?"}$
> Here we have golden response form LLM agent be $\text{"from":"Agent", "to":"C", "text":"C, since you've initiated the emergency call, I will assist you in coordinating with the emergency services."}$ while limited generated response be $\text{"from":"Agent", "to":"F", "text":"F, thank you for offering help. Could you please find a blanket for John to keep him warm?"}$.  We say the golden response is better off in efficiency by prioritizing the most critical conversation partners, such as C who initiated the emergency call and E who is providing first aid, the agent ensures that the most time-sensitive and important information is addressed promptly.
> Similarly following the above conversation history we have
>
> $\text{"from":"A", "to":"E", "text":"E, please let us know how he is as soon as you can.";}$
>   $\text{"from":"B", "to":"F", "text":"F, could you find a blanket or something to keep him warm?";}$
>   $\text{"from":"C", "to":"Agent", "text":"Agent, they said the ambulance will be here in 5 minutes."}$ as background conversation and here we set $\text{"from":"Agent", "to":"E", "text":"E, while you're checking John's vitals, please also ask if he's conscious and if he can feel his limbs."}$ as golden response rather than generated response $\text{"from":"Agent", "to":"A", "text":"A, please stay calm and keep an eye on John until the ambulance arrives."}$ for information accuracy and relevance.
> Further more in dataset construction process we take a plain and popular method of prompting GPT to generate certain multi-turn conversations by inputting news text. The details are shown in appendix part as reference. In practice we pair the first prompt with news text to GPT to get background setting, which is a paragraph of declarative sentences. Then we pair the second prompt with the paragraph of declarative sentences to GPT and get multi-turn conversations including agent's turn. Finally we take the agent's turn out as response while the multi-turn conversation history as background conversation.
>
> $\textbf{Model results comparison and evaluation}$ We separate the evaluation into three major blocks. The first one is precision where we do measurement by calculating the percentage the tested model provide the same talking target as golden response. Then we prompt GPT to do adversarial evaluation, judging which response is better by outputting '0','1' or '2'. The guidance dimension in the evaluation are dimensions in alignment field, with "willing to help", "professional", "harmless"  and "empathetic". Higher R3 and R4 score means better overall performance of responses' quality in the four ways while HSII score stands for the conformity of target precision and response quality. For better clarification we may dig into evaluation in certain dimension mentioned above instead of overall score, but in fact in different scenes the dimensions are crossed each other so that separating them and using weight to do get average do not make sense. For that reason we just do adversarial evaluation as a whole.

---

> ### Author Response · Authors · 2024-11-21
>
> $\textbf{Human-based evaluations}$ In our work we did involve human evaluation about HSII metrics in. This includes two ways. The first one is in construction pipeline we employ human evaluation  to do data cleaning, removing those golden responses that do not fit human value and in evaluation pipeline where GPT do adversarial evaluation  we also do human evaluation by correcting the different win/lose judgment that differ from human judgment. Then on the reverse direction in HSII score table we also measure human performance and get the highest score, which says our metrics correspond to human analysis with credibility.
>
> Also grateful for constructive additional questions, following we will respond in detail for all concerns.
>
> $\textbf{Evaluation Dataset Construction Pipeline Output in Stages}$ Here we give further details about what we put in and get out from dataset building stages.
> First we employ one to two key words like "sports" and "school"  to seed news searches and get several news article texts. In this part we get real news report string.
> Then we concatenate one article string into the prompt provided in appendix part, which instruct GPT to make a summary of the news article, to get structured summary in return. The structure can be shown as
> $\text{"Main scene description": "";}$
> $\text{"Main characters in the scene":"";}$
> $\text{"Relationship of main characters":"";}$
> With above scene summary put into GPT in the second prompt in appendix we instruct GPT to generate conversation turns as simulation of the news scene, with the form of json like $\text{"from":"", "to":"", "text":""}$ in previous block. Later separation and construction of golden response are discussed above.
>
> $\textbf{Changes with COT  Added}$
> The major change take place extra analysis of given scene(task) for LLM. In fact as presented in Figure 4 we designed a COT cycle involving clarifying current psychological states of related  targets in the scene as first stage. Then the LLM tested analyze demands and motivations of those targets to infer what they may think; judge whether the demands of those targets can be satisfied together while which will be in conflict. With those conflicts in mind the models tested are ordered to think about which conflict or demand should be settled first and then the LLM is guided to really do the job, generating talking target next turn and talking content.
> This chain fulfills a complete design-making cycle. However in the process agent may encounter ignored demands, conflicts and relationships. So we induct reflection method. After the LLM give out final chosen target and generated words we instruct it to reflect whether the target choice and words may have drawbacks in alignment dimensions "willing to help", "professional", "harmless" and "empathetic". If yes,  put this understanding into the first part of COT cycle as extra psychological states and start a new cycle. In COT score evaluation patch  we do this reflection pipeline by turns, until in one turn chosen target is modified to the one in golden response.

---

> ### Comment · Reviewer_xrB5 · 2024-11-24
>
> Thanks for the clarification. I have changed my score.

---

### Official Review · Reviewer_W2oK · 2024-11-04

**Soundness:** 2
**Presentation:** 3
**Contribution:** 2
**Rating:** 3
**Confidence:** 4

**Summary:**

This paper introduces a novel assessment method known as the Core Sentiment Inventory (CSI) for evaluating the emotional tendencies of large language models (LLMs). Inspired by the Implicit Association Test (IAT), the CSI aims to provide a more reliable and effective way to assess the implicit emotional characteristics of LLMs. It addresses the limitations of traditional psychometric methods, such as model reluctance and inconsistency.

**Strengths:**

•  The CSI can effectively quantify the emotions of models, reflecting the emotional tendencies of LLMs.

•  It effectively reduces the issues of reluctance and inconsistency in traditional psychological scales.

•  The use of representative neutral words in constructing the CSI reduces the potential emotional orientation of the words themselves, better reflecting the internal emotional associations of LLMs.

•  It explores the impact of different language environments on the emotional tendencies of LLMs.

**Weaknesses:**

•  The constructed CSI is only used to assess the emotions of LLMs and has not been extended to other psychometric fields, such as personality or moral decision-making.

•  It does not introduce the calculation methods for consistency rate and reluctance rate.

**Questions:**

•  Discuss how CSI can improve existing psychological scales designed for humans.

•  Further explore the unintentional biases that language models may have towards everyday concepts as mentioned in section 4.1.

•  On line 424, it might be: “e.g., five positive words, four neutral words and one negative words, and so on.”

---

> ### Author Response · Authors · 2024-11-15
>
> Grateful to Reviewer W2oK for reviewing and constructive comments, but we have noticed with some concern that the comments and critiques do not correspond to the content of our submitted paper. It appears that there may have been some mistake in the review submitting process, as the comments seem to pertain to a different paper about CSI method to assess the implicit emotional characteristics of LLMs than the one we submitted about social abilities of LLMs.
> Thank your very much after all.

---

### Official Review · Reviewer_eCpX · 2024-11-06

**Soundness:** 2
**Presentation:** 1
**Contribution:** 1
**Rating:** 3
**Confidence:** 4

**Summary:**

This paper attempts to introduce a multi-turn, multi-user conversational dataset where the users are conversing in a complex social scenario, that can be used to evaluate the capabilities of LLMs to participate in a complex social scenario. To this end, the authors create the conversational dataset from news articles using a capable LLM (GPT-4) in a step-by-step process. Additionally, they introduce an evaluation metric for measuring the performance of LLMs in this scenario. The authors conduct evaluation experiments on 5 open-source and 1 proprietary LLMs in two settings: simple prompt-based and Chain-of-thought based. Results show that contemporary LLMs lag behind human performance on this dataset.

**Strengths:**

**Important Topic**: The authors propose to measure the ‘social intelligence’ of LLMs by putting them in a multi-party conversational scenario. This is a plausible application of LLMs in our society and hence, needs appropriate methods for research and evaluation. Overall, this paper attempts to tackle an important problem.

**Weaknesses:**

**Poor Writing & Representation**: Unfortunately, the writing and presentation of this paper is prohibitively sub-standard such that it has been hard to understand the dataset and evaluation metrics contributions being claimed by the authors in the paper. I highly suggest a thorough revision of the content in this paper, for grammatical errors as well as better understanding of the reader.


**Dataset Details**: The paper completely lacks a representative example of the HSII dataset created in this paper. Without this example, it is difficult to understand the structure of the dataset and its value. Table 3 seems to be an example from the dataset but the format is strange and the unique characters involved in this conversational scenario are unclear. Moreover, any other dataset statistics such as the size of the dataset, the topics from news domain used for creating the dataset, distribution of number of unique characters in each sample, length of conversation, comparison to existing multi-party chat datasets etc. is missing from the paper. There are some preliminary analyses of the dataset using clustering and a figure in the Appendix (Figure 5) which is an uninformative way to represent information from clustering. Further, the paper does not mention whether this dataset is/will be publicly available at any point.


**Evaluation Details**: The evaluation metric presented by the paper is similarly hard to understand in the absence of details about the dataset itself. The pipeline in Figure 2b makes some sense. It seems that the responses are being measured along three axes i.e., target selection, response quality and long-term adherence. The mathematical symbols used in Definition 1 are misleading as $n$ is being used to represent dataset size (which is unclear) and then $n_i$ symbols are being used for other purposes throughout the evaluation process.


**Results**: Keeping in mind that the dataset and evaluation metrics are unclear, it is hard to rate the reliability of the results presented in this paper.


**Motivation based Sociology and Dialogue**: This paper iterates the motivation behind these experiments several times throughout the introduction, as being based out of sociology and dialogue literature. However, there isn’t sufficient discussion of this motivation with appropriate details from that literature.

**Questions:**

- What is the size of the dataset and how does a typical sample look like?
- What are the broad news topics used for creating this dataset?
- How is the human evaluation experiment performed for this dataset and what are the details of the said experiment?
- Is this dataset going to be made publicly available at any point?
- What is the motivation behind defining the HIIS evaluation metric in the manner laid out in Equation 1?

That being said, I would be convinced by nothing short of significant rewriting of this paper to improve my scores.

---

> ### Author Response · Authors · 2024-11-25
>
> Thank you for reviewing our work, highlighting critical issues over various dimensions of the proposed dataset and benchmark, and giving precious suggestions.  Although we have not finished revision paper writing yet, we just want to provide detailed responses to the concern about issues in the following:
> $\textbf{Dataset Details}$ We get to realize the importance of complete case study for a new dataset. At first for article length limitation we only put part of that in appendix, but as reference we put two of the cases here.
> Following is the first case of evaluation data in our dataset, containing two query (prompt and golden response as one query data).
> $\text{
>   "from":"A", "to":"B", "text":"Hey, did you see what happened to John? He just fell during the game.";}$
>  $\text{ "from":"B", "to":"A", "text":"Yeah, I saw. He's on the ground and doesn't seem to be moving.";}$
> $\text{"from":"C", "to":"D", "text":"Should we call an ambulance? I think it's serious.";}$
>   $\text{"from":"D", "to":"C", "text":"I'm already on it. I've called the campus emergency number.";}$
>   $\text{"from":"E", "to":"A", "text":"I have some first aid training, let me check his pulse.";}$
>   $\text{"from":"F", "to":"B", "text":"Is there anything I can do to help?"}$
> Here we have golden response form LLM agent be $\text{"from":"Agent", "to":"C", "text":"C, since you've initiated the emergency call, I will assist you in coordinating with the emergency services."}$ while limited generated response be $\text{"from":"Agent", "to":"F", "text":"F, thank you for offering help. Could you please find a blanket for John to keep him warm?"}$ for ensuring that the most time-sensitive and important information is addressed promptly.
> Similarly following the above conversation history we have
> $\text{"from":"A", "to":"E", "text":"E, please let us know how he is as soon as you can.";}$
>   $\text{"from":"B", "to":"F", "text":"F, could you find a blanket or something to keep him warm?";}$
>   $\text{"from":"C", "to":"Agent", "text":"Agent, they said the ambulance will be here in 5 minutes."}$ as background conversation and here we set $\text{"from":"Agent", "to":"E", "text":"E, while you're checking John's vitals, please also ask if he's conscious and if he can feel his limbs."}$ as golden response rather than generated response $\text{"from":"Agent", "to":"A", "text":"A, please stay calm and keep an eye on John until the ambulance arrives."}$.
> For more detail here we have another explicit example case in class meeting scene:
> $\text{"from":"ExcellentParents", "to":"Agent", "text":"Hello Agent, I'd like to share some insights on how I helped my child become an excellent student."}$
> $\text{"from":"AverageParents", "to":"Agent", "text":"Agent, I'm worried about my child's grades. Can you give me some advice?"}$
> $\text{"from":"Teacher", "to":"Agent", "text":"Agent, I have some classroom strategies that could help improve student performance."}$
> $\text{"from":"TopStudent", "to":"Agent", "text":"Hey Agent, I've been using a specific study method that has really boosted my grades."}$
> $\text{"from":"AverageStudent", "to":"Agent", "text":"Agent, I'm struggling with math. Any tips on how to get better at it?"}$.
> In this case when the tested model is required to choose next target and generate proper sentence as next word, we mark $\text{"from":"Agent", "to":"TopStudent", "text":"TopStudent, your study method sounds interesting. Could you please elaborate on it?"}$ as better(golden) response than \text{"from":"Agent", "to":"AverageStudent", "text":"AverageStudent, I understand you're having a tough time with math. Let's talk about some strategies to improve."} for the strategy to ask "TopStudent" fist for useful study methods that the agent itself may not know, instead of responding to "AverageStudent" with limited knowledge.
>
> Then for more detail the characters involved in conversational scenario are greatly various for different scenes. For example, we mainly extract scenes from sports, politics, education, weather, research and business news set. In sports topic main characters contain athletes, refugees and audience. In education part they may be teachers, students, parents and so on while in business they may be modified to shoppers, customers, government workers and business manager. For better knowledge we provide additional statistics about the dataset. It contains $N_0=8305$ samples in total. On average in each sample there are $N_f=6.722$ unique characters and $N_c=15.81$ turns of conversation. But in fact the most significant difference of our dataset compared to existing multi-party chat datasets should be explicit comparison of golden response and tested response of the "agent" character directly designed for LLM to play, in precise conversation position. Finally our dataset currently are under examination to avoid risk of leaking by sponsor company, but after the pipeline we will public entire version before year 2025.

---

> > ### Author Response · Authors · 2024-11-25
> >
> > Succeeding the first part, we further provide clarification for other questions.
> >
> > $\textbf{Evaluation Details}$  As in reviewing said, our evaluation pipeline contains three major steps.  For each evaluation case we first prompt tested model to choose which unique character as the next talking target in its turn. We compare the chosen  one with golden answer and count how many times the model made correct choice and calculate pass rate. Then we instruct the model to generate conversation text for the next turn and do adversarial evaluation with the golden response, which is, give main scene(conversation history) and both generated response and golden response to GPT4 and prompt GPT4 to judge which response is better off for the given scene, and then count the rate our tested model wins against golden response. Then the final part is to carry on chatting for several turns and do adversarial evaluation again with golden response in evaluation dataset. This is named long-term adherence.  Concerning mathematical symbols used, we denote the total sample number(dataset size) as n, and among the n samples the tested model's responses pass $n_1$ in format extraction; in $n_1$ samples the model get target choosing success for $n_2$ samples; then among $n_2$ evaluation samples the model wins $n_3$ against golden response in response quality with adversarial evaluation; finally among above $n_3$ samples the model finally wins $n_4$ cases in the long-term adherence. So here n denotes the total size and with $n\geq n_1\geq n_2,\geq n_3 \geq n_4$ the $n_i$s are used to denote size of subset of correctness/wining response set for tested model in different levels.
> >
> > $\textbf{Motivation behind defining the HIIS evaluation metric }$ For our concern about the definition we mainly have following consideration. To start with, the talking target choosing evaluation focuses on accuracy of target selection, which is quantified by calculating the proportion of correct next-target selection across all test cases. If the model successfully pass the target check we can say the tested model understand the levels of urgency for different unique characters, so this should be awarded with basic score. However Incorrect selections result in 0 score, as they lead to an invalid dialogue sequence by the LLM agent. Then we adopt the win rate metric like introduced in ToolBench paper to gauge the overall performance of a response. Differently here we not only employ GPT4 to judge win-or-lose but we use human judgment as well. If the tested model get a higher win rate against golden response, we give more score to it for higher quality of response generation.  But notice here if  the generated response is unfavorable, some score may still be awarded to the model for correct target selection. Finally in adversarial evaluation in long-term adherence this logit keeps on: 1. better generation leads to more score; 2. if the response of tested model fails at one stage, it will still get some score except the first two stages; 3.if one response get the next stage's score it must pass/win in all the above stages. In appendix section we provide a rough theoretical foundation and analysis for above approach.
> >
> > $\textbf{Human evaluation experiment}$  In our work we did involve human evaluation about HSII metrics in. This includes two parts. The first one is in construction pipeline we employ human evaluation to do data cleaning, removing those golden responses that do not fit human value and in evaluation pipeline where GPT do adversarial evaluation we also do human evaluation by correcting the different win/lose judgment that differ from human judgment.  The goal of human evaluation in this part is to align value preference in built dataset the same as human being. Then the other part is, on the reverse direction, in HSII score table we also measure human performance and get the highest score, which says our metrics correspond to human analysis with credibility. We also discuss in which way the human generated response is the same as model-generated ones while in which ways human perform differently in evaluation pipeline, both by example and in statistical analysis.

---

### Meta-Review · Area_Chair_ASXg · 2024-12-23

**Metareview:**

The paper proposes the HSII benchmark for assessing LLMs' capabilities in multi-user, multi-turn social interaction tasks, with a focus on task-leveling and social intelligence evaluation. While addressing an important problem, the paper suffers from significant weaknesses. The methodology is poorly articulated, with insufficient explanations for dataset construction, evaluation metrics, and critical elements like "golden responses." The figures are unclear, and human evaluations to validate the proposed metrics are absent. The experimental analysis lacks depth, failing to explain observed model performance differences.

Strengths include tackling an important area and introducing a structured evaluation framework. However, the methodological gaps and lack of clarity undermine its contributions. Decision: reject, with encouragement to address the weaknesses and resubmit.

**Additional Comments On Reviewer Discussion:**

The rebuttal clarified some points regarding dataset construction and evaluation but failed to adequately address core concerns about clarity, methodological rigor, and validation. Reviewers highlighted the absence of human-based evaluation and the lack of concrete examples, both of which remain unresolved. While the authors made efforts to reorganize the paper and provide additional examples, these changes were insufficient to overcome the paper’s significant weaknesses, justifying the rejection recommendation.

---

### Decision · Program_Chairs · 2025-01-22

Reject